# Chemical effects of diceCT staining protocols on fluid-preserved avian specimens

Catherine M. Early[1,2,3,4¤]*, Ashley C. Morhardt[5], Timothy P. Cleland[6], Christopher M. Milensky[2], Gwénaëlle M. Kavich[6], Helen F. James[2]

1 Biology Department, Science Museum of Minnesota, Saint Paul, MN, United States of America,
2 Department of Vertebrate Zoology, National Museum of Natural History, Smithsonian Institution, Washington, D.C., United States of America, 3 Department of Biological Sciences, Ohio University, Athens, OH, United States of America, 4 Florida Museum of Natural History, University of Florida, Gainesville, FL, United States of America, 5 Department of Neuroscience, Washington University School of Medicine in St. Louis, St. Louis, MO, United States of America, 6 Museum Conservation Institute, Smithsonian Institution, Washington, D.C., United States of America

¤ Current address: Biology Department, Science Museum of Minnesota, Saint Paul, MN, United States of America
* cmearly1311@gmail.com

**Data Availability Statement:** All proteomic data are available at ftp://massive.ucsd.edu/MSV000084970/. All scan data are available on MorphoSource (S3 Table; https://www.

## Abstract

Diffusible iodine-based contrast-enhanced computed tomography (diceCT) techniques allow visualization of soft tissues of fluid-preserved specimens in three dimensions without dissection or histology. Two popular diceCT stains, iodine-potassium iodide ($I_2KI$) dissolved in water and elemental iodine ($I_2$) dissolved in 100% ethanol (EtOH), yield striking results. Despite the widespread use of these stains in clinical and biological fields, the molecular mechanisms that result in color change and radiopacity attributed to iodine staining are poorly understood. Requests to apply these stains to anatomical specimens preserved in natural history museums are increasing, yet curators have little information about the potential for degradation of treated specimens. To assess the molecular effects of iodine staining on typical museum specimens, we compared the two popular stains and two relatively unexplored stains ($I_2KI$ in 70% EtOH, $I_2$ in 70% EtOH). House sparrows (*Passer domesticus*) were collected and preserved under uniform conditions following standard museum protocols, and each was then subjected to one of the stains. Results show that the three ethanol-based stains worked equally well (producing fully stained, life-like, publication quality scans) but in different timeframes (five, six, or eight weeks). The specimen in $I_2KI$ in water became degraded in physical condition, including developing flexible, demineralized bones. The ethanol-based methods also resulted in some demineralization but less than the water-based stain. The pH of the water-based stain was notably acidic compared to the water used as solvent in the stain. Our molecular analyses indicate that whereas none of the stains resulted in unacceptable levels of protein degradation, the bones of a specimen stained with $I_2KI$ in water demineralized throughout the staining process. We conclude that staining with $I_2KI$ or elemental $I_2$ in 70% EtOH can yield high-quality soft-tissue visualization in a timeframe that is similar to that of better-known iodine-based stains, with lower risk of negative impacts on specimen condition.

morphosource.org/Detail/ProjectDetail/Show/
project_id/978).

**Funding:** CME: National Science Foundation (NSF)
Graduate Research Fellowship Program (GRFP)
(https://www.nsfgrfp.org/) under NSF DGE
1060934 and DGE 1645419 and the NSF GRFP
Graduate Research Internship Program. HFJ, CME,
ACM: Alexander Wetmore Fund of the Smithsonian
National Museum of Natural History Division of
Birds (https://naturalhistory.si.edu/research/
vertebrate-zoology/birds). TPC, GMK:
Smithsonian's Museum Conservation Institute
(https://www.si.edu/mci/) Federal and Trust Funds.
The funders had no role in study design, data
collection and analysis, decision to publish, or
preparation of the manuscript.

**Competing interests:** The authors have declared
that no competing interests exist.

# Introduction

Diffusible iodine-based contrast-enhanced computed tomography (diceCT; www.diceCT.
com) [1] has emerged recently as an exciting and powerful technique for visualizing in situ,
cadaveric soft tissues in animals. DiceCT allows specimens to be "virtually dissected," which
may eliminate the need for traditionally destructive dissection techniques (e.g., gross dissec-
tion) [1]. However, even the "non-destructive" diceCT method may alter specimens in other,
unknown or untested ways. Natural history museum curators are increasingly called upon to
evaluate requests for loans of fluid-preserved anatomical specimens for diceCT [2,3], without
much information about how the specimen's physical or chemical condition may be altered by
the procedure. Lack of evidence-based standards for diceCT staining may lead to unnecessarily
conservative policies for specimen loans or conversely, unintentional damage to rare and/or
delicate specimens.

Certain deleterious diceCT artifacts have already been observed and occasionally mitigated,
such as specimen shrinkage and deformation [1,3–8] and specimen demineralization [9,10].
However, the underlying mechanistic causes of some of these artifacts—and of the desired dif-
ferential staining and increased radiopacity of tissues attributed to diceCT—are not well
understood due to the challenges of studying iodine solutions [11]. Studies that have aimed to
mitigate diceCT artifacts have focused almost exclusively on matching solute concentration of
stains to solute levels in living tissues (i.e., "physiologically isotonic") [4,5], whereas no studies
have specifically considered the potential effects of various diceCT solvents on museum speci-
mens (but see [12]).

Popular diceCT stain solvents include (1) deionized water (e.g. [4,13]), and (2) 100% EtOH
[12,14,15]. In water (a polar solvent) at high pH, $I_2KI$ dissociates into $K^+$ and $I_3^-$ ions, (with
some $I_3^-$ ions eventually reacting to become $I_2$ and $I^-$), and all ions potentially react with water
molecules to produce other chemical species [16]. In contrast, in 100% EtOH (a less polar sol-
vent than water), elemental iodine ($I_2$) equilibrates with alcohols through the formation of
"outer" and "inner" complexes [16–18]. Conventional solution chemistry states that, in both
cases (water and 100% EtOH), triiodide ions ($I_3^-$) form ($I_2 + I^- \rightarrow I_3^-$), giving each solution a
dark brown or black color. Triiodide, specifically, has been shown to stain animal soft tissues a
reddish-brown color [19] by interacting with glycogen in the tissues [1,20]. Given the equilib-
rium $I_3^- \rightarrow I_2 + I^-$, molecular iodine ($I_2$) is present in stain solution and diffuses into the speci-
men. Within the specimen, it is reduced, provoking an increase of serum iodide. Therefore,
visible staining, though caused by the charged species $I_3^-$, also gives rise to real incorporation
of iodide ($I^-$) into tissues at the molecular level [16].

The specifics of how iodine binds to glycogen have been the subject of much study [21–23],
but there appears to be little detailed information in the literature that links how initial applica-
tion of triiodide induces formation of glycogen-iodine complexes. Also, the specifics of how
triiodide interacts at the molecular level with other components of animal soft tissues are not
well understood. Finally, within the diceCT literature, it is stated that triiodide binds to soft tis-
sues, making them radiopaque [1]; however, as this mechanism has yet to be verified at the
molecular level, it remains unclear if triiodide is the singular "active ingredient" responsible
for both visual (i.e., brown color) and CT contrast-staining of all components of soft tissues.

Both water- and ethanol-based stains are widely available and proven effective from over a
century of use, especially as histological stains [16]. However, the diffusion-based nature of
diceCT raises new concerns regarding the use of these two stains, especially in cases where
large and/or whole-body specimens must be immersed in stain for hours or even months. For
instance, long-term storage in water is known to degrade soft tissues [24,25], and exposure to
100% ethanol is known to dehydrate and significantly shrink specimens [3]. Given that most

museum fluid specimens are stored indefinitely in 70% EtOH, placing a museum fluid specimen directly into either an aqueous or 100% EtOH solvent may produce unwanted artifacts. Treating such specimens with an iodine-based stain mixed in 70% EtOH could eliminate the impacts of changing the concentration or type of solvent in which it is stored, but the effectiveness of staining in 70% EtOH has not yet been evaluated.

Here, we compared two popular stains used in diceCT protocols, Lugol's iodine ($I_2KI$ in water; [26]) and $I_2$ in 100% EtOH, with two relatively untested iodine stains mixed in 70% EtOH. *Passer domesticus* (house sparrow) specimens (conspecifics, prepared identically, stained independently) were assessed for quality of soft-tissue visibility on CT scans of stained specimens, effects of stain solvents on specimen condition, and modifications of proteins from muscle and bone tissues. In the process of comparing the stains, we identified a potentially overlooked mechanism of iodine binding to specimen tissues. Our specimens stained with water-based stain also underwent an unanticipated change to their physical condition. The results of our study expand our mechanistic understanding of diceCT and provide guidance on choosing a stain solvent for curatorial staff and practitioners of diceCT alike.

## Materials and methods

### Specimen collection

This study was carried out in strict accordance with the recommendations in the Guidelines for the Use of Wild Birds in Research by the Ornithological Council. The protocol was approved by the Animal Care and Use Committee of the Smithsonian National Museum of Natural History (NMNH) (Protocol Number: 2017–13). All subjects were euthanized quickly and humanely via cardiopulmonary compression, a method that minimizes suffering, before being subjected to study.

We netted adult house sparrows (*Passer domesticus*) on 2 Nov 2017 and prepared them as specimens immediately after euthanasia. We washed the specimens with soapy water to reduce water repellency due to natural oils, and using a syringe and small gauge needle, injected 10% neutral-buffered formalin (NBF) into the chest, legs, wings, neck, and lower body cavity of each specimen. We submerged the specimens in NBF for 3 days, rinsed them with water, and then transferred them into graded EtOH concentrations of 25%, 50% and 70% at intervals of three to seven days, following the protocol recommended by [25] and following the staining rationale of [27].

### Staining and scanning

One specimen was placed in each of the stains in Table 1. Typically, diceCT uses either water (deionized or reverse osmosis (RO)) or 100% EtOH for staining solvent [1]. Here, we developed two additional stains using 70% EtOH as a solvent. We selected specimens with similar body masses to ameliorate the influences of size on the rate of staining.

All specimens were preserved in 70% EtOH, so the specimen stained with $I_2$ in 100% EtOH was dehydrated stepwise (incubated in 70%, 80%, 90%, 100% EtOH for 48 hr intervals) following established protocols [25,27,28] before staining. We avoided refreshing stains as this could influence staining quality, but refreshing the Lugol's iodine stain was necessary as the color of the stain lightened throughout the staining process, which may indicate a decrease in stain concentration (e.g. [29]). All four specimens were placed in their respective stains at the same time and scanned once a week throughout the staining process with a GE phoenix v|tome|x m microCT scanner at a resolution of 65–72 µm. The scans were examined with phoenix datos|x as they were generated to assess the progress of the staining process. Because of unexpected changes in curatorial condition of the specimen in Lugol's iodine, a second specimen was stained in water-based stain with an earlier stain refresh time (USNM 657965; Table 1). Each

**Table 1. Stain type and staining duration.**

| Specimen Number | Mass (g) | Stain | Weeks in Stain |
|---|---|---|---|
| USNM Birds 657964 | 36.1 | 3.75% $I_2KI$ in RO water (Lugol's iodine), refreshed after 7 weeks in stain | 10 |
| USNM Birds 657965 | 36.4 | 3.75% $I_2KI$ in RO water (Lugol's iodine), refreshed after 4 weeks in stain | 6 |
| USNM Birds 657968 | 33.2 | 1.25% $I_2$ in 100% EtOH | 6 |
| USNM Birds 657963 | 38.7 | 3.75% $I_2KI$ in 70% EtOH | 5 |
| USNM Birds 657967 | 36.4 | 1.25% $I_2$ in 70% EtOH | 8 |

The specimen number, specimen mass immediately before staining, concentration and formula of the stains used on each specimen, and the duration of staining for each specimen. Specimens were scanned prior to staining and weekly during the staining process to assess progress.

specimen was left in stain until its soft tissues were stained throughout, at which point the specimen was removed from stain and placed in 70% EtOH for diffusion-based de-staining. The specimen stained with $I_2$ in 100% EtOH was rehydrated to 70% EtOH from 100% EtOH in successively lower concentrations of EtOH as part of the de-staining process.

As soft tissues generally visualize with poor contrast on CT scans, specimens were considered to be fully stained when the soft tissues in the center of the abdomen (farthest region from the surrounding stain fluid) were well-resolved and could be differentiated on the scans. Degree of staining did not necessarily need to be equivalent across all tissues for a specimen to be considered fully stained because iodine binds to tissues differentially, because some amount of heterogeneous staining is expected with a diffusion-based system, and because beam hardening artifacts may be present in the data. Similar standards were used to visually assess the quality of staining on CT scans. Specifically, (1) visibility of different soft tissue types, (2) clarity of detail of soft-tissue structures of the muscles, lungs, and digestive tract, (3) degree of staining of external tissue, such as the beak compared to tissues deep in the abdomen, and (4) the degree of staining of soft vs. hard tissues were used to define quality of staining.

## Curatorial assessments of specimen condition

We assessed the physical condition of the specimens after the staining and scanning were completed and the specimens had been destaining in 70% EtOH for at least 148 days. To detect physical signs of demineralization of bone or deterioration of soft tissues, we: (1) grasped the ends of long bones (ulna, and if available, tarsometatarsus) and gently attempted to bend them, (2) pressed on the dorsal braincase to assess firmness, and (3) pressed on the breast muscles, thorax, and lower abdomen to assess whether they felt firm or soft. These observations were made in comparison with a control specimen that had been collected, fixed, and preserved in 70% ethanol on the same date, from the same locality, and in the same manner as the experimental specimens. We also tracked the specimens' body masses throughout the staining process as a measure of shrinkage.

## Testing pH of water-based stain

Differences in the physical condition of the water-based stained specimens prompted pH tests of water-based stains roughly six months after the end of the staining process. The pH of the

Lugol's iodine solution used on USNM 657964 and the RO water used as the solvent in this stain was measured with an Oakton pHTestr 20 (Cole-Parmer, Vernon Hills, IL, USA). The pH of extra Lugol's iodine solution that was mixed for this experiment but never used on any of the specimens was also tested. The Lugol's iodine solution used to stain USNM 657965 had been discarded and was not available for pH testing. An additional experiment was performed on a batch of Lugol's iodine made with the same protocol as the experimental stains. The pH of the RO water as well as the pH of this new stain were measured at one minute, two days, and five days after mixing.

## Sample preparation for proteomics

Resected tissue samples of bone and muscle were taken from the most distal locations available on the tibiotarsus to avoid compromising the integrity of the specimen's body cavity. Bone samples were scraped clean with a scalpel. Contralateral samples were taken minimally at two different time points—first, after formalin fixation and preservation in EtOH, and second, after staining. For specimens that needed their stain solutions refreshed before they became fully stained, samples were also taken before refreshing the stain solution. Samples were then stored at -40°C until analysis.

## Proteomics

Proteins from bone and muscle were extracted as follows in brief (detailed proteomic methods in S1 File): 1) bone samples were homogenized in 400 mM ammonium phosphate dibasic, 200 mM ammonium bicarbonate, 4M guanidine HCl [30] and 2) muscle proteins were homogenized sequentially in 50 mM ammonium bicarbonate then 0.5% SDS (final concentration). 10 μg of protein from each extract (bone and muscle) was taken, single-step reduced and alkylated [31], and digested with modified trypsin (Promega) using the modified single-pot solid-phase sample preparation (SP3) method [32,33]. Peptides were desalted using C18 stage tips [34]. Subsequently, 0.5 μg of peptides were separated on in-house packed Thermo BioBasic C18 nanoliquid chromatography columns and detected on an LTQ Orbitrap Velos (Thermo-Scientific) mass spectrometer.

Resulting RAW files were searched with either PEAKS 8.5 or MetaMorpheus 0.0.301 [35] against a *Taeniopygia guttata* database. Both search algorithms included iodination of histidine (H) and tyrosine (Y) (PEAKS: iodo (HY) and diiodo (HY); MetaMorpheus: iodo (HY), diiodo (HY), triiodo (Y)) as possible modifications. For PEAKS data, peptide spectral matches (PSMs) were filtered at a 1% false discovery rate (FDR). For MetaMorpheus, PSMs and proteins were filtered at 1% FDR for each level. Full parameters of each algorithm are included in S1 File.

## Quantification of tyrosine, histidine, and total iodination

A custom R script (see S2 File) was written to calculate the levels of iodination (tyrosine [Y]/histidine [H]), diiodination (Y/H), and triiodination (Y) on the PSM level as quantified by MetaMorpheus. These PSM level evaluations were then combined based on protein accession to give protein level iodination levels. An additional calculation of peptide level iodination was performed using a 1000 replicate bootstrap method. This provided a calculated mean, standard deviation, and confidence interval for each staining condition. The calculation methodology is derived from [36].

## ATR-FTIR method

To avoid destructive sampling (e.g., using TGA/DSC tests) of limited bone samples, we used ATR-FTIR to calculate mineral-to-matrix ratios. Bone samples for attenuated total reflection

(ATR)-Fourier transform infrared spectroscopy (FTIR) were placed directly on a Golden Gate ATR (diamond crystal, single bounce, 45˚) accessory coupled to a Thermo Nicolet 6700 Fourier transform infrared (FTIR) spectrometer with deuterated triglycine sulfate (DTGS) detector to evaluate changes in mineralization. A total of 64 scans were taken for each bone sample with a resolution of 4 cm$^{-1}$. A piece of aluminum foil was used to back the sapphire anvil to eliminate any sapphire absorption in the IR spectrum. FTIR spectra were identified using the Infrared and Raman Users Group (IRUG) libraries, the HR Hummel Polymer and Additives library, and the ASTER mineral library.

All baseline and ratio calculations were performed using an automated program in TQAnalyst EZ version 8 (Thermo Scientific). The amide I peak at 1647 cm$^{-1}$ was baseline corrected from 1712–1575 cm$^{-1}$, and the integrated area of the phosphate v1 and v3 of $PO_4^{-3}$ (1200–800 cm$^{-1}$) was baseline corrected from 1200–800 cm$^{-1}$ [37–40].

## Results

### Contrast-enhanced CT scans

A total of 30 scans were taken before all specimens were deemed fully stained (e.g., Fig 1), and all scans are available on MorphoSource (S3 Table). Staining duration of each specimen depended on the stain used (Table 1). The quality of contrast on CT images of each fully stained specimen also varied by stain (Fig 2). On CT scans, all specimens seemed to exhibit a diffusion-based staining pattern, with superficial tissues appearing more brightly stained relative to deeper tissues. Specimens subjected to water-based stains show a greater difference in staining brightness between more superficial tissues versus deeper ones (Fig 2A)—a phenomenon sometimes referred to as the "rind effect." Additionally, soft tissues in the specimens subjected to water-based stain are much more radiopaque than bones and other mineralized materials (e.g., stones in the crop), to the point that the bones are hardly visible on the diceCT scans of these specimens (Fig 2A). In contrast, the bones of all three specimens treated with ethanol-based stains remained radiopaque when specimens were fully stained (Fig 2B–2D). CT images of the specimen stained in $I_2KI$ in 70% EtOH appear grainy compared to the other two ethanol-based stain specimens but still allow tissues and organs to be differentiated from each other (Fig 2C).

### Changes in specimen condition

No differences in physical condition between any of the specimens were noted at the start of the experiment aside from the minimal differences in body mass (Table 1). Body masses of the three ethanol-based stained specimens decreased initially by a modest amount (Fig 3). The water-based stained specimen gained mass by the end of the staining period after having undergone an overall decrease in mass for most of the staining period.

At the end of the experiment, while taking samples from the birds for chemical analyses, we noted detrimental changes in the physical condition of two of the specimens. For the bird that was stained in 100% ethanol (USNM 657968), we noted that the skin and muscle samples felt hard and brittle and that the feathers on the skin sample dried very quickly when removed from solution. This is evidence of dehydration, which is to be expected when specimens are moved from 70% to 100% EtOH [28]. For the bird that was stained in $I_2KI$ in RO water for ten weeks (USNM 657964), we noted loose contour feathers in the rinse water combined with softness and flexibility in the tibiotarsus sample, which dried to a dark color. The bone sample taken from this specimen before refreshing the stain had been hard and dried to a whitish color. We also found that this bird had developed a very flexible tarsometatarsus (Fig 4) and that its breast and thorax were soft to the touch.

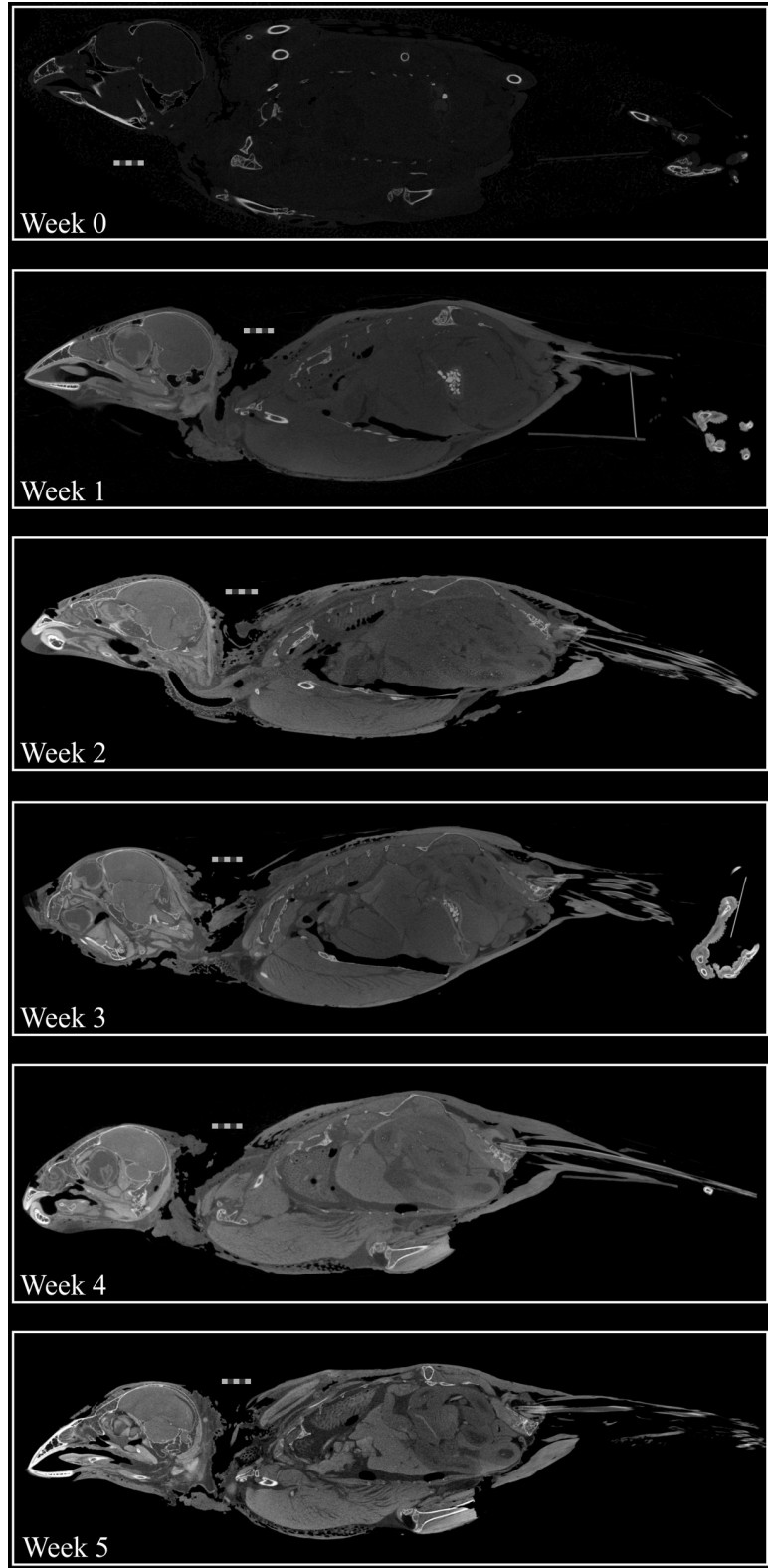

**Fig 1. Stain progression over five weeks.** Progressive staining of USNM 657963, the specimen stained with $I_2$KI in 70% EtOH. The stain can be seen to penetrate deeper into the specimen with every week of staining. Scale bars = 5 mm.

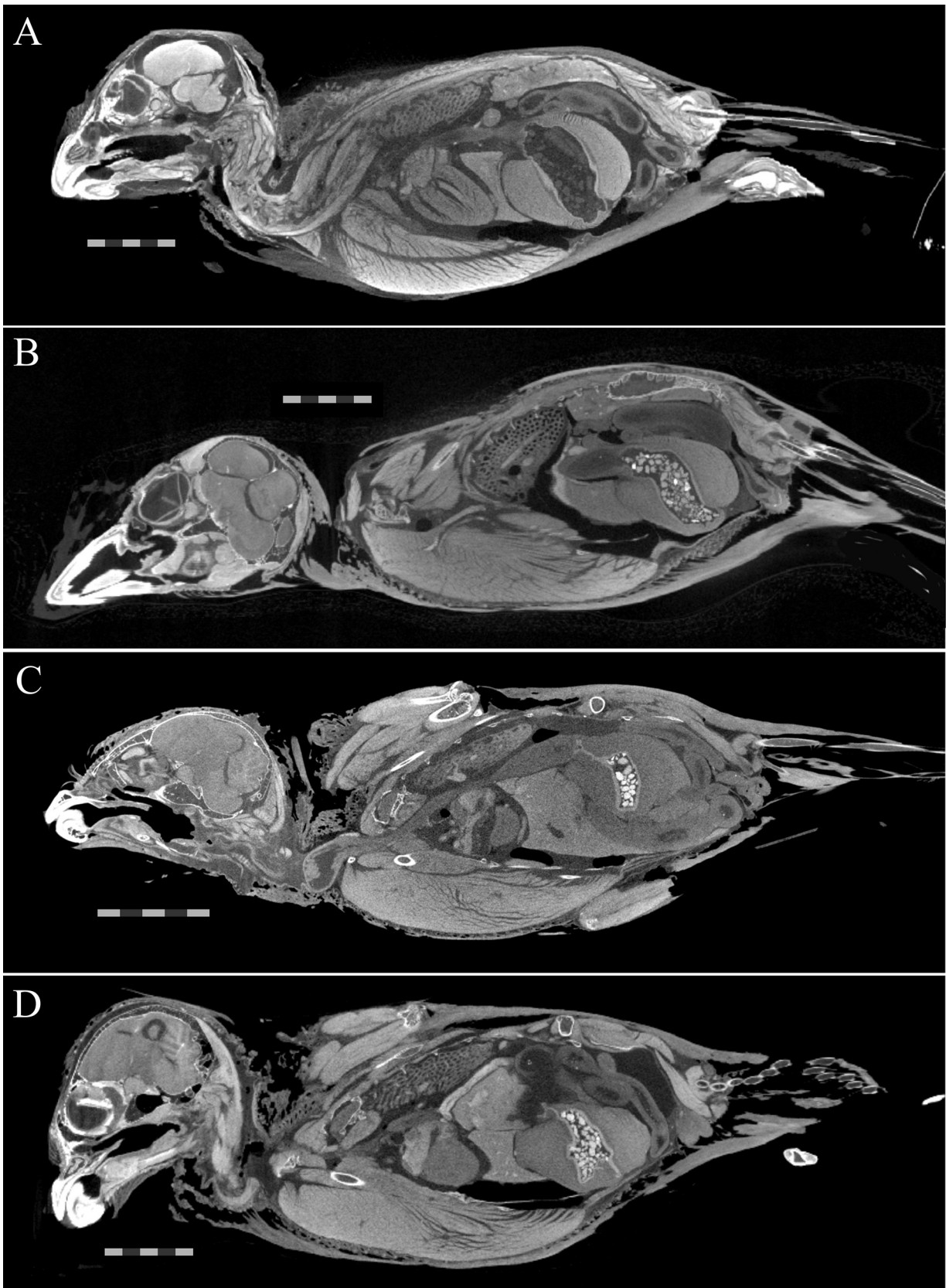

**Fig 2. diceCT scans of house sparrow specimens.** Slice images from CT scans of iodine-stained specimens that were generated when each specimen was considered "fully stained" based on visual assessment of the scans. A) USNM 657964 after ten weeks in $I_2KI$ in water. B) USNM 657968 after six weeks in $I_2$ in 100% EtOH. C) USNM 657963 after five weeks in $I_2KI$ in 70% EtOH. D) USNM 657967 after eight weeks in $I_2$ in 70% EtOH. Scale bars = 1 cm.

We assessed the physical condition of the specimens after they had been destaining in 70% ethanol for 148 to 342 days. Our assessments highlighted the demineralization of bone and potential maceration of soft tissues that took place in the two birds stained in a water-based solution (Table 2). In both specimens stained in Lugol's iodine, we observed the skull, breast, wings and legs to be notably soft and flexible in comparison with the other experimental specimens and with typical formalin-fixed, fluid-preserved specimens. We also could not feel the sternum by pressing on the abdomen ventrocaudally in these two specimens, whereas we could in the other specimens.

## Acidity of water-based stain

The pH of the RO water at the NMNH was 7.1. After using this water to mix stain, the pH of the stain was 6.4 within one minute, 4.25 after two days, and 3.74 after five days. Water-based

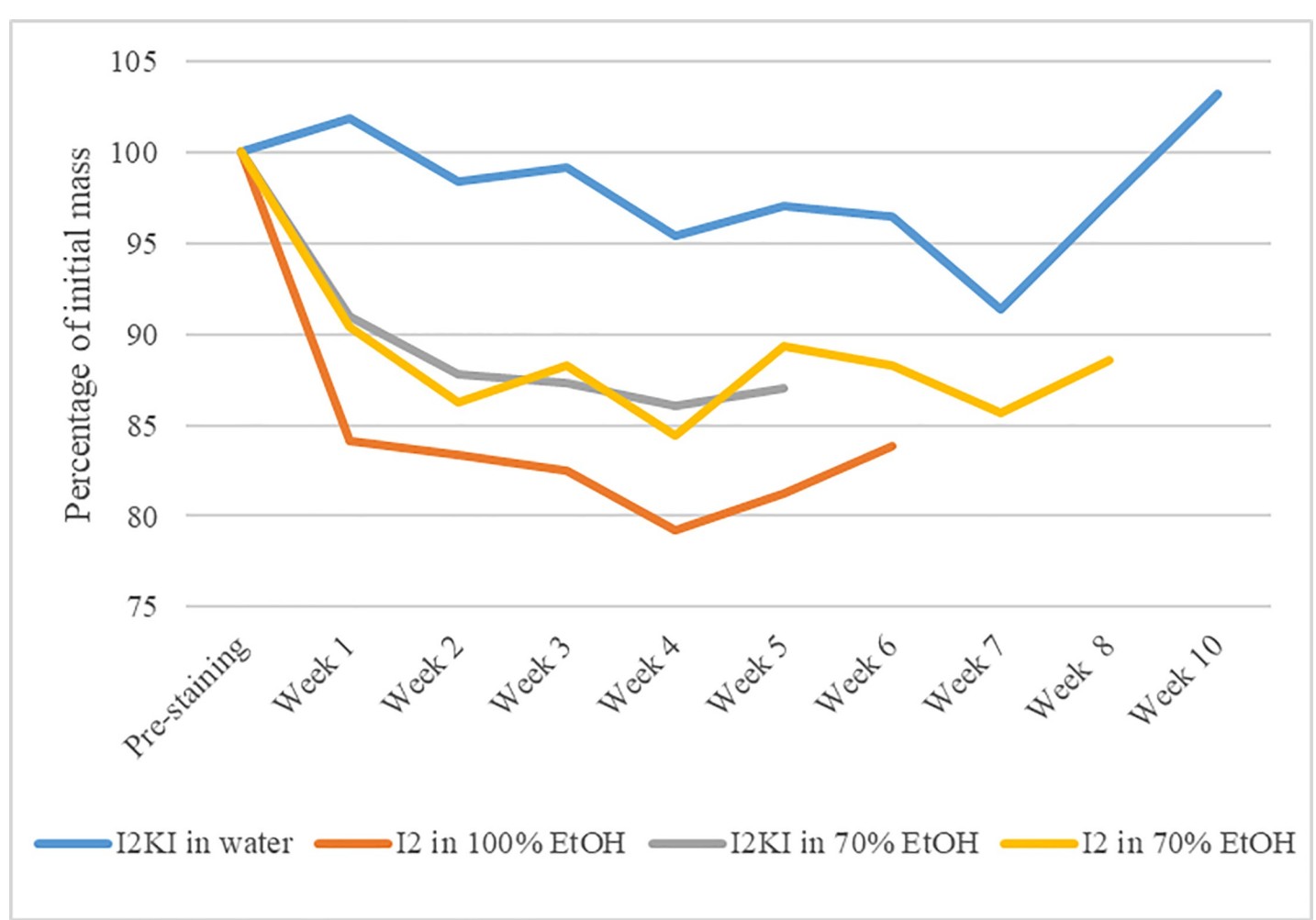

**Fig 3. Body mass through time.** Body masses of the specimens treated with different iodine stains throughout the staining process.

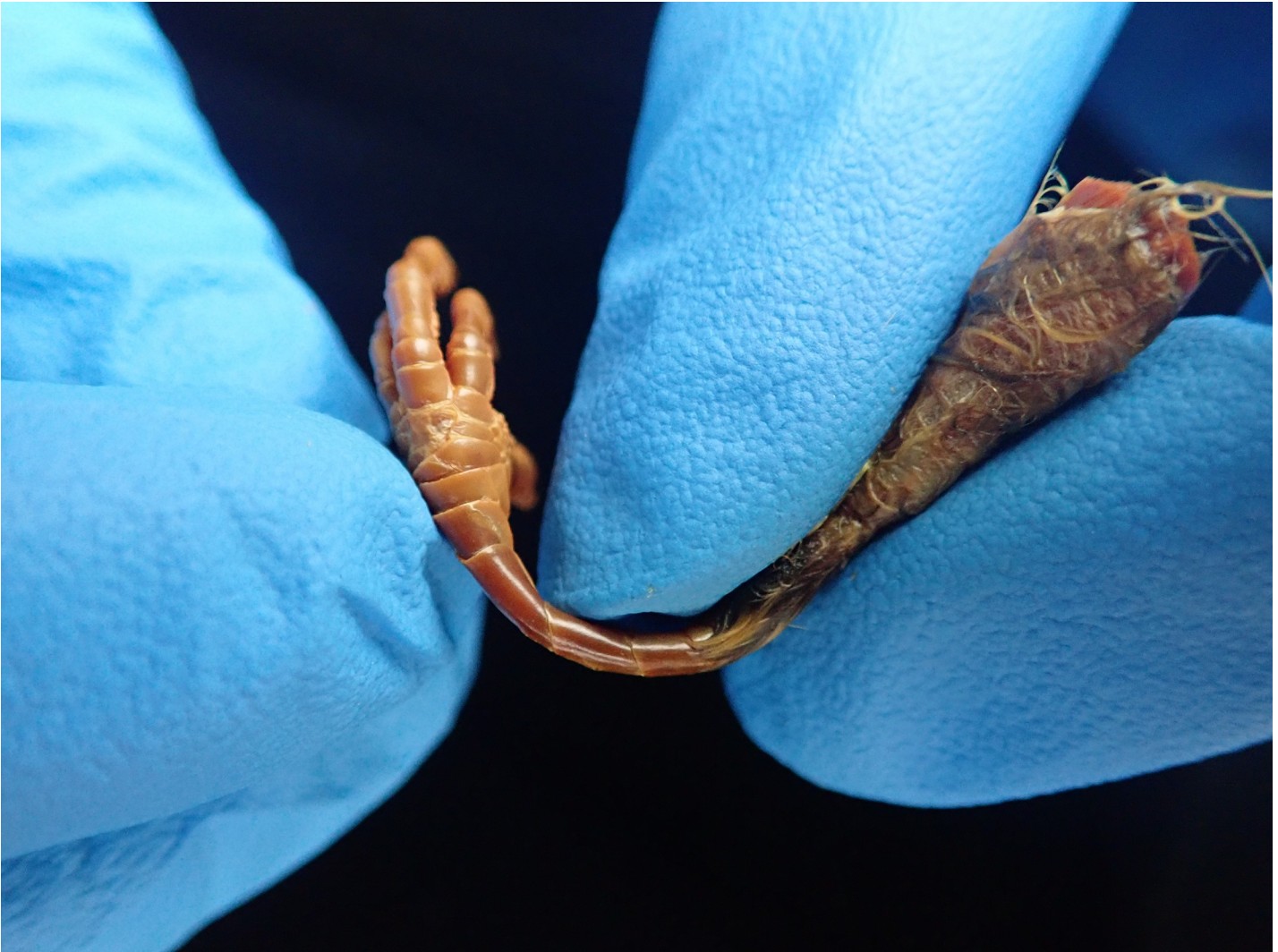

**Fig 4. Demineralized leg.** The leg of USNM 657964 after ten weeks in $I_2KI$ in water. The tarsometatarsus is being bent with very little effort, showing demineralization of the bone.

stain in which the first specimen stained with Lugol's iodine (USNM 657964) soaked for seven weeks was measured as pH 3.20. The stain that was used to refresh this specimen and interacted with the specimen for three weeks until it was fully stained was measured as pH 2.83. All of these values for water-based stains used at varying stages of the experiment indicate

**Table 2. Assessment of the physical condition of specimens after staining.**

| Stain | Specimen | Days Staining | Days De-staining | Long Bones | Cranial Vault | Soft Tissues |
|---|---|---|---|---|---|---|
| **control** | | 0 | 0 | **stiff** | **hard** | **firm** |
| $I_2KI$ in water | USNM Birds 657964 | 82 | 319 | flexible | soft | soft |
| $I_2KI$ in water | USNM Birds 657965 | 76 | 148 | flexible | soft | soft |
| $I_2$ in 100% EtOH | USNM Birds 647968 | 46 | 337 | stiff | hard | firm |
| $I_2KI$ in 70% EtOH | USNM Birds 657963 | 41 | 342 | stiff | hard | firm |
| $I_2$ in 70% EtOH | USNM Birds 657967 | 60 | 323 | stiff | hard | firm |

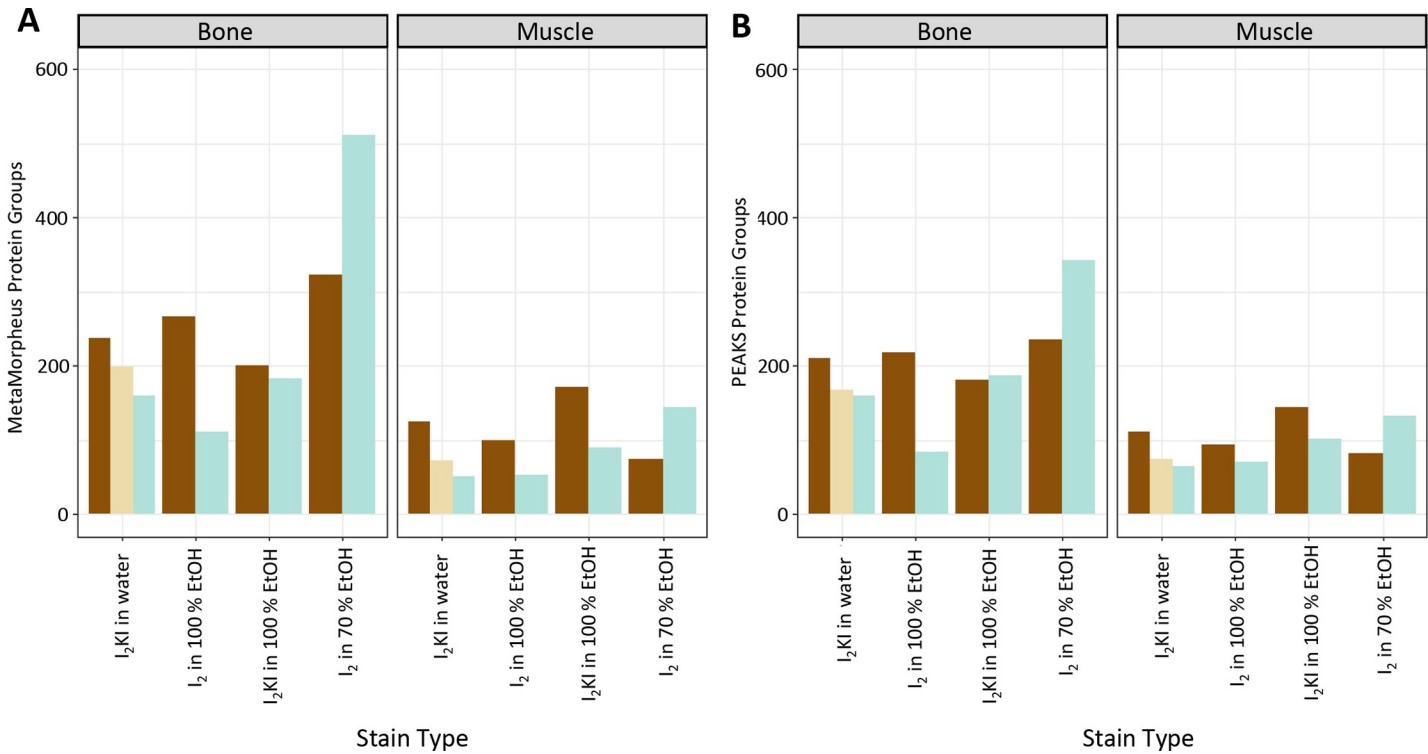

**Fig 5. Protein groups.** MetaMorpheus (A) and PEAKS (B) protein groups for bone and muscle samples. $I_2$KI in water had an additional mid-stain sample during the stain refresh. Pre- and post-staining samples are indicated by dark brown and teal bars, respectively. Samples taken during staining from the specimen stained with $I_2$KI in water are indicated with tan bars.

relatively high acidity. Acidity of EtOH is notoriously difficult to assess [41], so pH values of all other stains in our study could not be measured for comparison.

## Proteomics

Large numbers of proteins (Fig 5) were detected for all of the prestaining and iodine-stained conditions. Generally, fewer protein groups were detected post-staining, with the exception of $I_2$ in 70% EtOH, where more protein groups were detected in both bone and muscle. For the water-based stain, a progressive reduction in protein number was detected for both bone and muscle. This same trend is observed in samples from the $I_2$ in 100% EtOH stain specimen and in the muscle samples from the specimen stained in $I_2$KI in 70% EtOH. The bone samples from that same specimen have a similar number of proteins before and after staining.

## Iodination evaluation

All staining conditions, regardless of solvent, resulted in some level of iodination (Figs 6 and 7). The highest detected iodination occurred on the interim- and post-stained muscle samples from the water-based stain specimen (63.2% ± 7.1% and 64.0 ± 6.3% of both Y and H iodinated, respectively). The water-based stain resulted in the highest levels of iodination (76.5% ± 3.8%) for bone as well. The lowest level of staining in bone was in the specimen stained in $I_2$KI in 70% EtOH (18.8% ± 3.1%) and the lowest in muscle was in the specimen stained in $I_2$ in 100% EtOH (1.8% ± 0.9%). The specimen stained in $I_2$KI in 70% EtOH was the second lowest for muscle with similar levels (20.3% ± 3.5%) to its bone modification levels.

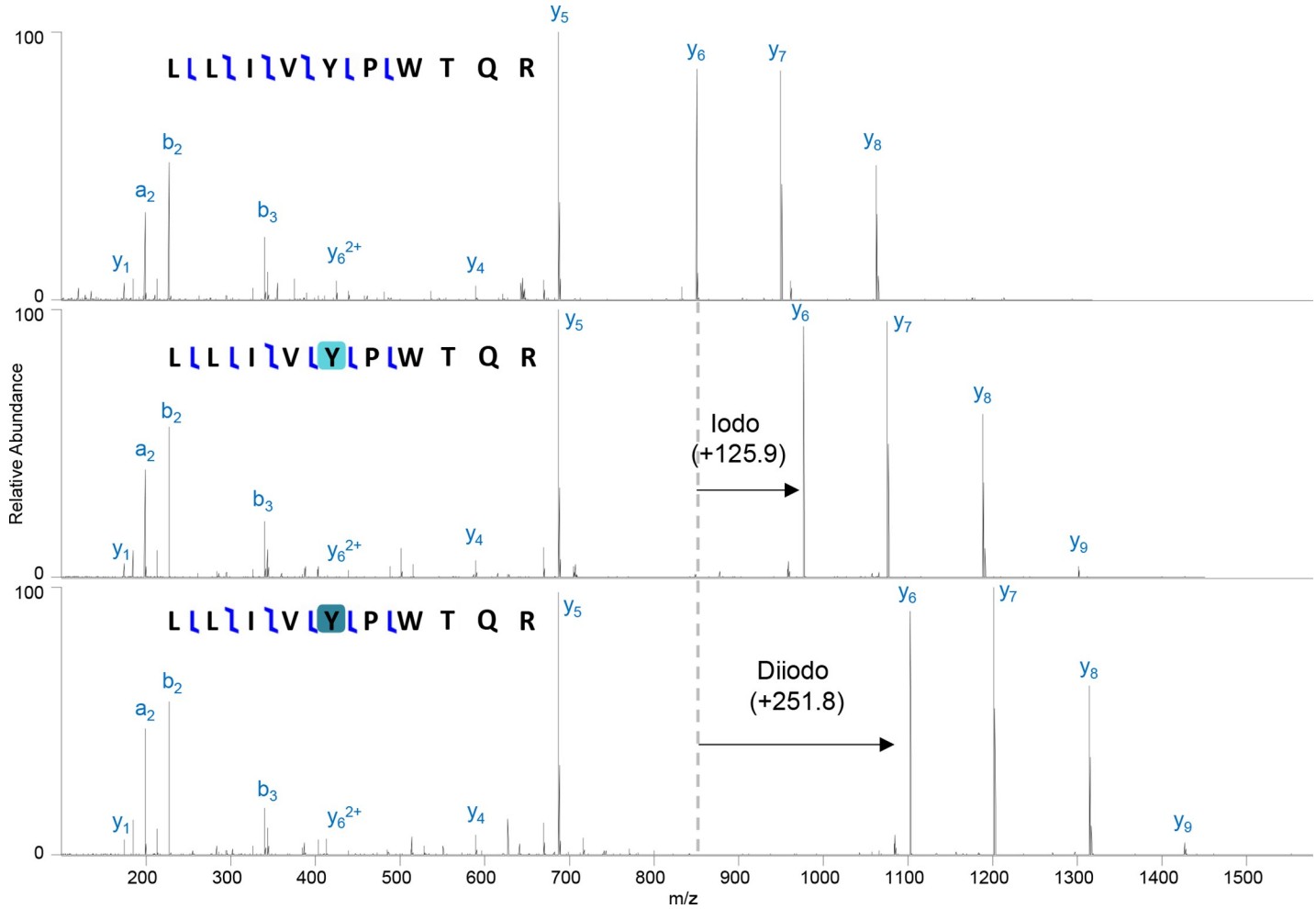

**Fig 6. Iodination of hemoglobin.** Hemoglobin beta subunit peptide (LLIVYPWTQR) showing unmodified (top), iodination (middle), and di-iodination (bottom) on tyrosine. Iodination is indicated with light blue and di-iodination is indicated with teal. Fragments $y_6$, $y_7$, $y_8$, $y_9$ show the mass shift corresponding to the addition of iodine(s).

Either collagen I was not found to have any iodination, or the peptides containing modifiable amino acids went undetected, likely because of the limited number of tyrosines and histidines present in the protein (for comparison in chicken because the collagen I sequence is unavailable for *Passer domesticus*: 12 Y, 14 H of 3152 amino acids). Therefore, all of the detected iodination on the proteins from the bone sample are derived from non-collagenous proteins (e.g., hemoglobin [Fig 6]). More specifically, in the bone samples, cytoskeletal (e.g., various actins, tubulins, myosins, alpha actinin), nuclear (i.e., various histones), residual muscle (i.e., myoglobin), and blood-derived (i.e., serum albumin, hemoglobin alpha and beta) proteins all showed iodination. Similar modified proteins were detected in the muscle extracts including myoglobin, actin, myosin, tropomyosin, albumin, hemoglobin, and histones.

## FTIR results

Four samples of selected bones from the second specimen subjected to water-based staining (USNM 657965), one bone sample from each of the three ethanol-stained specimens (USNM 657964, USNM 657967, USNM 657968), and one bone sample for each of two control birds

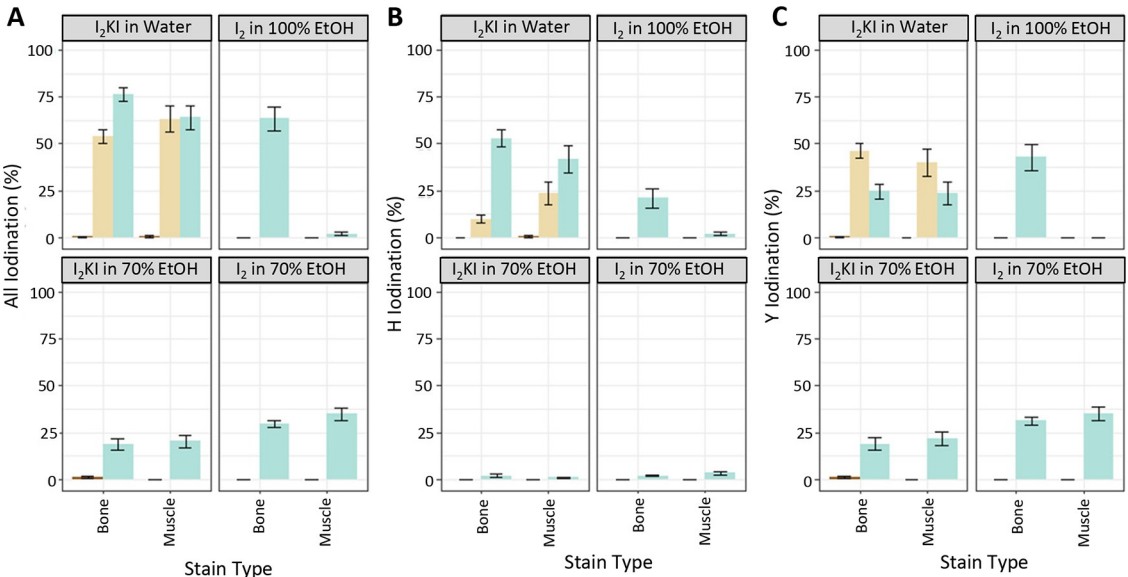

**Fig 7. Percentage iodination of muscle and bone.** Total iodination (A), H iodination (B), and Y iodination (C) percentages calculated from 1000 bootstraps of peptide level iodination for both bone and muscle. Pre- and post-staining indicated by dark brown and teal bars, respectively. Interim staining of the $I_2KI$ in water specimen is indicated with tan bars.

(USNM 657966, USNM 657969) underwent ATR-FTIR analysis to detect possible demineralization caused by the iodine-based staining. The ATR-FTIR spectrum in region 1200–850 cm$^{-1}$ is usually referred to as mineralized bands ν1 and ν3 of stretching phosphate ($PO_4^{-3}$), and the 1800–1200 cm$^{-1}$ region determines the organic content. Demineralization is evidenced by the progressive reduction of the phosphate bands with respect to the amide I band of the collagen (assuming that the collagen is unaffected by the staining process).

Mineral content was measured by calculating the mineral-to-matrix ratio—that is, the ratio of the integrated area of the phosphate ν1 and ν3 of $PO_4^{-3}$ (1200–800 cm$^{-1}$) to the amide I (1647 cm$^{-1}$) calculated as the absorbance area under the band between (1712–1575 cm$^{-1}$) as shown in (Fig 8). All staining types resulted in some degree of demineralization (Table 3, S1 Fig) where the mineralization-to-matrix ratio decreases after staining, with the maximum reduction in ratio observed in the water-based stain. The results of the mineral-to-matrix ratios within the Lugol's iodine stained individual exhibit a decrease from the unstained bone (sample 657965AB), to the partially-stained one (657965ZB), and to the fully-stained one (657965BB) (Table 4, Fig 8). The value obtained for the sample recovered from the destaining sample (657965B-X) is within the range of the one fully stained (Table 4). While a decrease in protein group number was observed for bone stained with Lugol's iodine, the most abundant bone protein (i.e., collagen I) was still detected and not iodinated, so the mineral-to-matrix ratio likely reflects the actual decrease in mineral content between the unstained and post-stained samples.

## Discussion

Our experiments allow direct comparisons of various iodine-based stain solvents in terms of the quality of staining, the impacts of the stains on the physical condition of the specimens, and the molecular effects of the stains. Proteomic analyses indicated that protein recovery rates were variable among the different stains but generally high enough in all stains to suggest that staining with iodine does not result in significant protein degradation and thus may not

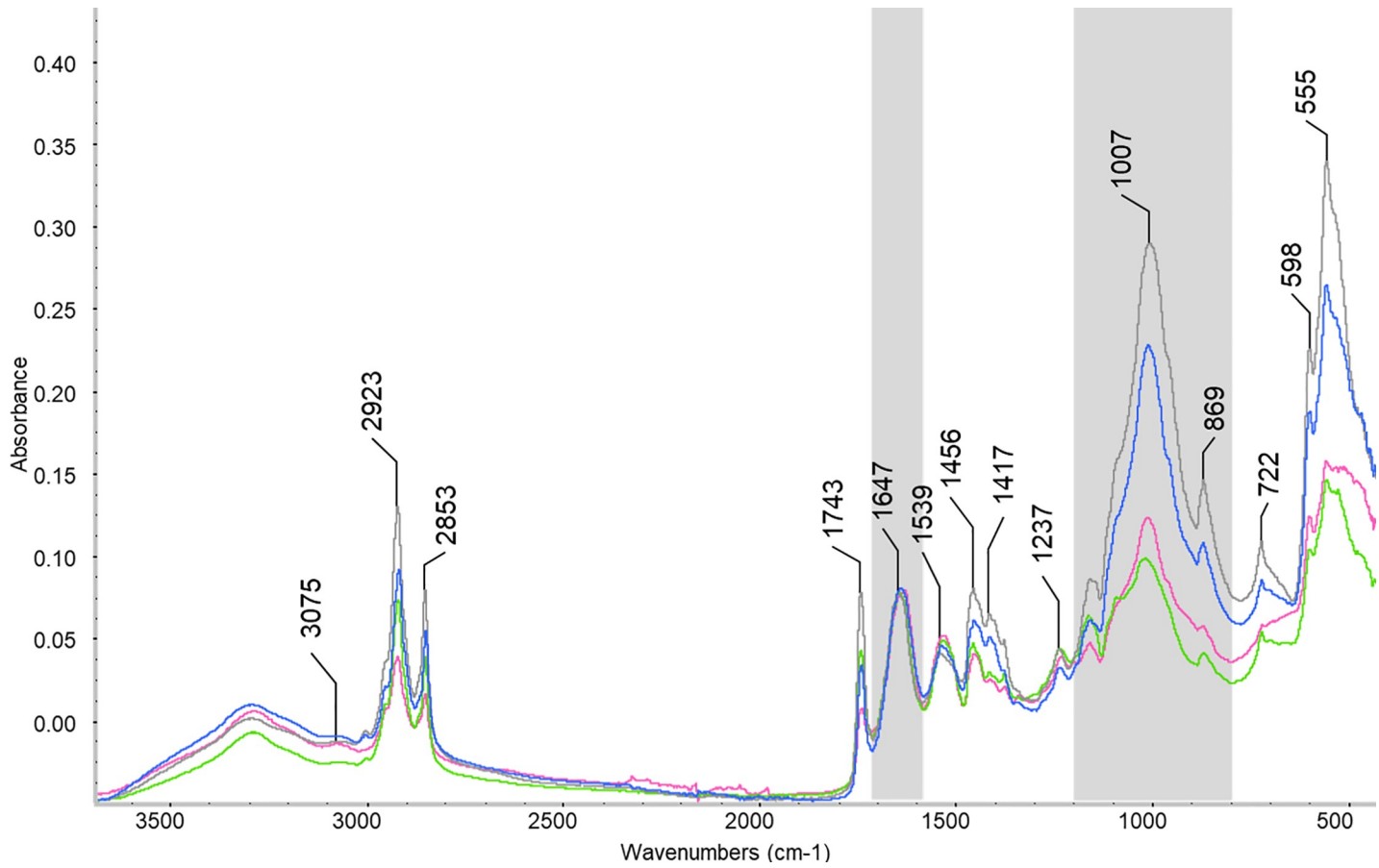

**Fig 8. Demineralization of bone of specimen stained in Lugol's iodine.** Overlay of the ATR-FTIR spectra of $I_2KI$ in water stained bird bone (657965). Prestained (gray), interim sample (blue), post-stain (light green), and de-stained (pink). The phosphate ν1 and ν3 of $PO_4^{-3}$ region is shaded in gray. Normalized to the Amide I peak from collagen at 1647 cm$^{-1}$ (also shaded in gray).

account for much of the observed degradation of specimens stained with Lugol's iodine. Pervasive demineralization of specimens subjected to water-based stain was observed in physical condition assessments of the specimens and in ATR-FTIR analyses of bone samples; these stain solutions were acidic and their acidity increased during staining. Our mass spectrometry results indicate that binding of iodide to proteins is mediated by iodination of amino acids, a previously undescribed molecular mechanism of diceCT. Interestingly, this mechanism is implicated in staining across different solvent types. We found that both of the stains made

**Table 3. Mineral-to-matrix ratios obtained from ATR-FTIR analysis of control and post-stained specimens.**

| Samples | Description | Mineral-to-matrix ratio |
|---|---|---|
| 657966YB | Control (unstained) | 8.36 |
| 657969YB | Control (unstained) | 6.37 |
| 657963YB | Stain 3: 1.25% $I_2$ (via 3.75% $I_2KI$) in 70% EtOH | 5.77 |
| 657968YB | Stain 2: 1.25% $I_2$ in 100% EtOH (I2E) | 4.92 |
| 657967YB | Stain 4: 1.25% $I_2$ in 70% EtOH (I2E) | 4.10 |
| 657965BB | Stain 1: 3.75% $I_2KI$ in RO water (Lugol's iodine) | 2.57 |

Table is ordered by decreasing Mineral-to-matrix ratio.

**Table 4. Mineral-to-matrix ratios obtained from ATR-FTIR analysis of a specimen stained with Lugol's iodine before, during, and after staining.**

| Sample | Description | Mineral-to-matrix ratio |
|---|---|---|
| 657965AB | Not stained | 9.27 |
| 657965ZB | Moderately stained | 6.13 |
| 657965BB | Fully stained | 2.57 |
| 657965B-X | Post-staining | 2.82 |

with 70% EtOH yielded high-quality staining of soft tissues comparable to that yielded by more commonly used stains, albeit in a somewhat longer time frame. The quality of staining may be partially correlated with iodination levels of proteins in bone and muscle, which varied between solvent types. This study presents direct data on the impacts of iodine stains on the proteins and mineralized tissues of fluid-preserved avian specimens. It also provides experience-based recommendations for curatorial staff considering requests to subject their specimens to these protocols, and to researchers using these protocols on museum specimens.

## Physical condition of specimens

The pattern of changes in mass in the three specimens stained in ethanol-based stained solutions is similar to the pattern of changes in volume observed by [5], with an initial decrease followed by leveling out (Fig 3). The mass of the specimen stained in Lugol's iodine was slightly higher by the end of the stain after decreasing throughout staining (Fig 3), which might be related to the changes in the physical condition observed in this specimen. Compared with an unstained specimen and with the specimens stained in EtOH, the soft tissues of the water-based stained specimens were less firm, the skull vault was soft and flexible, the sternum could no longer be felt by pressing on the trunk, and the long bones of the wing and leg were extremely bendable, especially in the specimen that was stained for ten weeks (Fig 4, Table 2). The softness of the skull vault, loss of a hard sternum, and flexibility of the long bones of the two specimens stained in water-based solutions, after destaining, are physical indications that demineralization had occurred.

## Protein degradation

Evaluating the number of protein groups detected before and after staining can be used as a proxy for the amount of protein degradation and/or protein leaching. In most cases, fewer protein groups were detected after staining, suggesting that proteins are broken down or lost during the incubation period (Fig 5). This is especially evident in the water-based stain, where progressively fewer protein groups were detected across the refreshed stain in both bone and muscle. This decrease in protein group number is also quite pronounced in the specimen stained with $I_2$ in 100% EtOH. It remains undetermined why the samples in 70% EtOH have a similar number of proteins in bone before and after staining for the $I_2KI$ stain and much greater numbers of bone proteins after treatment with the $I_2$-only stain. The increases in identifications could be derived from stochastic differences in data acquisition, but future studies with additional samples will clarify if this increase in the number of protein groups is intrinsic to staining with $I_2$ in 70% EtOH. The similar protein number before and after staining suggests that the $I_2KI$ in 70% ethanol has limited impact on the extractability of the bone protein samples, but the higher number of proteins after staining with $I_2$ in 70% ethanol suggests that this stain increases the solubility of the bone proteins. A similar, yet smaller magnitude, increase in

protein count in the muscle was also observed for the $I_2$ in 70% ethanol. Future studies will investigate the mechanism for why more proteins were detected post-staining.

Despite some loss of protein group numbers for some of the stains, proteins were still detectable in all post-stained samples, suggesting that staining up to the 10 weeks tested here does not fully destroy the tissues, with the caveat being that these proteins are heavily modified by iodination. Long-term studies are required to evaluate how iodination changes the stability of fluid preserved proteins.

## Demineralization of bones

Via μCT, we observed equal or lower levels of radiopacity of bone compared to those of soft tissues in the specimens stained in water-based solutions (Fig 2A, S1 Fig), despite the fact that bone proteins were equally or slightly more iodinated than muscle proteins in this specimen (Fig 7). Even if the proteins of the two tissue types were equally iodinated, the inherent radiodensity of mineralized tissue (e.g., bone) compared to unmineralized tissue (e.g., muscle) should still result in bones appearing more radiopaque than muscles, which can be observed in the two specimens stained in 70% EtOH (Figs 2C, 2D and 7). The fact that the specimens treated with water-based stain show the opposite pattern of radiopacity of their tissues than expected based on levels of protein iodination suggests that the bones of this specimen lost radiodensity via demineralization throughout the staining process. An alternative hypothesis that we cannot reject is that the disparity in radiopacity of soft tissues and bone in the water-based stained specimens is due to other radio-dense iodide species interacting with other tissue types. Our CT scans did not include a sample of material of known density across scans, so grayscale values of bones throughout the staining process were not directly compared to confirm that their absolute, rather than relative, radiopacity decreased. A different study used μCT to document evidence of progressive demineralization of bone of adult rats that had been stained in Lugol's iodine [10].

Although there are many examples of staining vertebrate specimens in Lugol's iodine without apparent demineralization based on physical condition of their specimens [1], we observed extensive demineralization of our avian specimens after staining in water-based stain. In ATR-FTIR analyses of bone samples from one of the specimens, USNM 657965, we observed a reduced phosphate area when compared to the amide I peak (Fig 8), which confirms that demineralization of bone occurred in the water-based stained specimen that was stained for six weeks. Bone samples from ethanol-based stains also showed some demineralization (S1 Fig; Table 4), but to a much lesser extent than the water-based stain. Care must be taken when using Lugol's iodine to perform diceCT to prevent or reduce demineralization during the process because it generates low pH solutions that can directly demineralize the bone tissue. Demineralization may be especially pronounced in specimens with thin bone cortices such as birds. It may be more limited in species with thicker/more dense bone cortices because the acid cannot access as deeply within the bone in the relatively short staining time. Future investigations of buffered water-based iodine staining may be warranted to develop protocols that reduce the demineralization process.

## Acidity of Lugol's iodine

Notably, water-based stain solutions used in this study began as slightly acidic (pH = 6.4) when first mixed and increased in acidity (pH = 2.4) over the course of staining. Additionally, it appears that water-based stain acidity increased over time and with specimen exposure. These results mirror those of [10], and they make sense in light of what is known about the solution chemistry of iodine in water, as well as how iodide binds to biomolecules. For

instance, free iodine ($I_2$) does not readily dissolve in water. Instead, it reacts with water molecules per the following equation:

$$I_2 + H_2O \rightarrow HOI + H^+ + I^-$$

where HOI (hypoiodous acid) and dissolved hydroiodic acid ($H^+$ and $I^-$) contribute to acidity. In initially neutral or acidic conditions, this reaction runs slowly, accounting for both the initial acidity of stain, as well as at least some of the gradual increase of acidity over time [42,43]. This scenario is exacerbated by the presence of undissolved iodine crystals remaining out of solution for an extended period of time [43]. Therefore, ways to help mitigate the acidification effects of $I_2$ in water are: (1) adding small, additional amounts of KI when mixing up stain solution until all $I_2$ crystals are fully dissolved [16], (2) replacing used stain with newly-mixed stain as often as possible during the staining of a specimen, and (3) not mixing stain or stock stain solution prior to the day of an experiment. Failure to mitigate the acidification of stain likely was the cause of demineralization of bone in our specimens. Thus, we urge caution.

Additionally, our results show that exposure to specimen tissues also seems to acidify the water-based stain solutions. Given that the interaction between stain solution and tissues appears to result in the iodination of certain biomolecules, and provided that this interaction has been correctly characterized as an eventual substitution of $I^-$ for $H^+$ ions within the tissues and surrounding fluid via oxidation/reduction reactions [16], it is possible that the resulting $H^+$ ions released from the stained tissues then contribute to the acidity of the surrounding stain fluid. It is yet unclear how this process may be altered to mitigate acidification, but likely the addition of a buffer solution and/or other excipients to the system could prove effective in the future. However, given that specimen tissues appear prone to shrinkage in response to increases in solute concentration (e.g., [4,5,7]), it is important that any additional solutes not then lead to excessive hypertonicity of the stain relative to the specimen's internal environment (e.g., serum/plasma levels).

pH testing is inapplicable to EtOH-based solutions [41], so we have no data on the acidity of the stain solutions for specimens stained in EtOH. Based on the FTIR data, we do observe demineralization of bone in these solutions, suggesting that pH is an issue in EtOH-based stains as well, albeit to a much lesser degree than the in water-based stain. Future research will be necessary to evaluate methods (like those described above for water-based stains) to minimize demineralization in EtOH-based stains.

## Iodination of proteins as staining mechanism

Radiopacity of soft tissues in diceCT has historically been hypothesized to be derived from $I_3^-$ binding to the tissue (e.g., [1]); however, here we found no evidence for this type of binding to proteins. Instead, we detect sequential iodination (i.e., $I^-$ replacing hydrogens one at a time [up to three binding]; Fig 6) of tyrosines and histidines. During the iodine staining process, iodide is added to tyrosine (Y) and histidine (H) through replacement of hydrogens, with iodine on the phenolic and imidazole groups, respectively [44–46]. This addition can occur as a single iodine (iodination, +125.896648 Da) on H or Y, two iodines (di-iodination, +251.793296 Da) on H or Y, or three iodines (tri-iodination, +377.689944 Da) on Y only (Fig 6). We hypothesize that this iodination is one of the primary mechanisms for radiopacity of soft tissues in diceCT and explains why predominantly proteinaceous tissues (e.g., feathers, eye lens) become highly radiopaque in this process. The observed iodination results in similar chemical structures to those used in small molecular contrast agents used in medical CT (e.g., single iodines bound to phenolic rings [47]), further supporting our hypothesized radiopacity mechanism.

All four staining methods resulted in iodination of histidine and tyrosine to varying levels (Figs 6 and 7), which likely drove the observed increases in radiopacity of soft tissues. Many different proteins were iodinated (e.g., hemoglobin, Fig 6), but we were unable to detect iodination on collagen I in bone. This likely reflects limited numbers of histidine and tyrosines (we detected none) in collagen I. It could also indicate that these regions of collagen I become more insoluble or are lost during the iodine treatment. Most of the collagen I tyrosines occur near the terminal ends of the triple helix, so may be more available for loss. Despite this, the bones of our specimens still became iodinated via the iodination of the many non-collagenous proteins present in this tissue.

The amino acid modifications observed in the muscles and bones of our specimens may be irreversible or only partially reversible because the iodine covalently binds to the tyrosine and histidine residues, which may lead to permanent changes to fluid-preserved specimens stained in iodine (e.g., increased radiopacity of soft tissues, changes to the molecular composition of the specimen). Future research will investigate the levels of reversibility of this iodination. Additionally, other organic molecules (e.g., lipids, polysaccharides) need to be characterized in a similar manner to evaluate whether iodination modifies them as well.

## Potential chemical basis of visual staining and contrast enhancement

Visual staining (i.e., color change observed visually) of glycogen by triiodide has been studied and explained previously. The exact physics behind the color change are not well understood, though they may involve the formation of polyiodide chains [48]. It is also known that once triiodide diffuses into tissues, free molecular iodine ($I_2$) can be formed spontaneously from triiodide, be reduced in the body to $I^-$, and then bound to organic molecules like amino acids and fatty acids [16,47]. Iodine, iodide, and triiodide are all radiopaque due to their high density. As stain solution diffuses into a specimen, a visual stain "front" of color change is observed as triiodide progresses gradually to the specimen's core. This progression is mirrored by the stain front on CT images. In the wake of the front, persistent, differential staining on CT images is likely the product of how much or how little iodide binds to various organic molecules present in the tissues, including amino acids. It is likely that diceCT staining yields tissues that are both saturated in triiodide, resulting in the color change in the specimen visible to the naked eye, and bound to iodide, resulting in enhanced contrast in CT scans.

## Quality of differential staining

Similar to the specimen stained in 100% EtOH in this experiment (Fig 2B) and those of [12], the bones of the specimens stained in 70% EtOH were more radiopaque than the surrounding soft tissues and were still visible on fully-stained scans (Fig 2C and 2D). The fact that the pattern observed in specimens stained in 100% EtOH (this study; [12]) was replicated in our specimens stained in 70% EtOH suggests that it is tied to solvent type. For the specimen stained in 100% EtOH, the higher radiopacity of the bone compared to the soft tissues may also be driven by the fact that the bones of this specimen were more iodinated than the muscles (Fig 7). The fact that the bones were more radiopaque than the muscles of specimens stained in 70% EtOH can likely be attributed to the fact that unstained bones are already more radiopaque than unstained muscles as these specimens experienced almost equal iodination of bone and muscle (Fig 7).

In contrast, by the time the soft tissues of the water-based stain specimens were fully stained, the soft tissues were much more radiopaque than the bones, making it difficult to visualize hard and soft tissues on the same CT scan (Fig 2A). This pattern is particularly interesting in light of our finding that the bone amino acids were slightly more iodinated than muscle

amino acids in the first water-based stain specimen and that we observed the highest degree of demineralization of bone in a water-based stained specimen (Figs 4 and 7, Table 4). These relative levels of radiopacity (greater in muscle than bone) mirror those observed in many other specimens stained with Lugol's iodine [1], which suggests that demineralization in water-based stain is not unique to our experiment.

Both of the stains mixed in 70% EtOH ($I_2$, $I_2KI$) yielded high-quality staining in a matter of just a few weeks longer than stains mixed in water or 100% EtOH (Fig 2, Table 1). The scans of the specimen stained in $I_2KI$ in 70% EtOH (Fig 2C) appear to yield grainier images than those of the specimen stained in $I_2$ in 70% EtOH (Fig 2D), but we cannot identify a mechanism to explain this difference. Despite our best efforts to get the specimen treated with $I_2$ in 70% EtOH to be fully stained, some of the intestines in the center of the abdomen appear to be only faintly stained (Fig 2D). This limited penetration to the center of the specimen and longer staining time is a drawback to the $I_2$ in 70% EtOH stain when compared to others in our experiment. One possible explanation for this is that, because 70% EtOH is not polar enough to fully dissociate all $I_2$ molecules without the addition of KI, the $I_2$ in 70% EtOH stain was of lower iodine concentration than the other stains used in our experiment. This problem could be circumvented by adding a known amount of more solute to an iodine stain made with 70% EtOH, keeping in mind that it would be difficult to be certain of the concentration of the stain itself. Future studies should attempt to improve penetration of $I_2$ in 70% EtOH to the center of specimens through higher stain concentration, longer staining times, or refreshing of the solution throughout staining.

## General recommendations for staining museum specimens

Curators are accustomed to evaluating requests for destructive study of fluid-preserved anatomical specimens, which are collected with the understanding that they may be dissected to document gross anatomy or sectioned for histological studies. In the case of traditional dissection, only the researcher performing the dissection interacts directly with the anatomy of the specimen, whereas diceCT enables any number of researchers to study minute anatomical details of the same specimen. In this sense, diceCT can be considered less destructive than dissection and can open doors to broader participation in comparative anatomy. However, in terms of having an unaltered anatomical specimen to curate after staining, our results and the work of others has shown that diceCT cannot be considered a non-destructive, fully reversible technique. Below, we offer some advice to curators and researchers aimed at encouraging diceCT for museum specimens while conserving the specimens for future work.

Because our chemical analysis indicates that iodine becomes bound to amino acids in the specimen during staining, we predict that soft tissues will remain more radiopaque than they were originally, despite "de-staining." Thus, the detailed models of the skeleton that can be readily obtained from CT scans of unstained anatomical specimens may become difficult to generate after staining. To maximize the information obtained from rare specimens, it would be wise to routinely scan those specimens before staining, to obtain data from the skeleton while it is still decidedly more radiopaque than the soft tissues.

Staining in Lugol's iodine caused demineralization of bone in our test specimens, likely because the Lugol's stain solution had a low pH. Further research is needed to determine how widespread this problem is. We note that the reduction in radiopacity observed in bones of specimens stained in Lugol's iodine in other studies that specifically looked at bone radiopacity suggests that it is not restricted to our experiment [10,12]. The two specimens we stained in the Lugol's solution also exhibited softening or maceration of soft tissues. We did not detect a clear signal of this change in our analysis of muscle proteins, although it was very evident in

our assessment of the physical condition of the specimens. We suggest that the extended time period in acidic, water-based stain is leading to these changes. We also hypothesize that the degree of fixation, which was complete and standardized in our specimens but is known to vary in archived museum specimens, may influence the susceptibility of a specimen to these changes.

For archived museum specimens, we propose staining with $I_2$ or $I_2$KI in 70% EtOH as an alternative to staining with Lugol's iodine. These stain solutions produced high quality scans in a reasonable time, and they allow specimens to be stained in a solution with the same ethanol concentration that is employed for their long-term storage. Although we cannot speak to the acidity of ethanol-based stains, our data do show that the specimens we stained in ethanol were less demineralized than those stained in Lugol's iodine. We acknowledge that this is a relatively untested approach compared with water-based iodine staining, so it could have other risk factors of which we are unaware.

Any specimen subjected to iodine-based stain, regardless of solvent, will inevitably need to be de-stained. Visual destaining using chemicals is fairly straightforward, with triiodide (brown) being converted to iodide (colorless) in the presence of sodium thiosulfate [1,49]. Additionally, our chemical analysis suggests it is unlikely that all iodine species are completely unbound from tissues during de-staining, even though the specimen may lose its brown coloration. These ions may continue to interact with the specimen and surrounding fluid in specimen jars after staining, and thus continue to cause the types of changes to the molecular condition of specimens that we have documented in this study. We therefore recommend that previously stained and then de-stained specimens should be stored in their own containers.

In view of our results, we warn curators of the possibility that they could be left with a demineralized and softened specimen after staining in Lugol's iodine. Although ethanol-based iodine stains may ameliorate these concerns, the proteins of any specimen stained with iodine are likely permanently iodinated. When granting permission for diceCT, it would be reasonable to ask the researcher borrowing the specimen to provide a good quality CT scan of the specimen prior to staining to digitize the skeleton and, if staining with Lugol's iodine, to track the pH and the physical condition of the specimen throughout the staining process. To mitigate acidic conditions during staining with Lugol's iodine, researchers are urged to: (1) use excess potassium iodide when mixing stain to ensure that all iodine crystals dissolve as quickly as possible without the opportunity to react with the surrounding aqueous solvent, (2) not mix up stain or stock solutions prior to experiments, and (3) refresh stain solution as frequently as possible. In addition, information about the staining and de-staining process should be documented and cataloged as part of the specimen's metadata, to guide future curation. These data can inform decisions about whether these specimens are good candidates for scanning with other imaging modalities or for staining with other chemicals, which should only be attempted with an understanding of how those chemicals will react with any remaining iodine.

## Summary of findings

Our results shed new light on the molecular mechanisms and long-term impacts of iodine-based staining of formalin-fixed, ethanol-preserved avian specimens. We identified sequential iodination of proteins as an important contributor to the radiopacity of soft tissues in diceCT, which has been an overlooked mechanism in the existing diceCT literature. We characterized differences in iodination levels and degradation of muscle and bone proteins of our specimens across different solvent types. We demonstrated progressively higher acidity of water-based iodine stains through time and in complement to those results found notable levels of demineralization of the bones of specimens treated with these solutions. Of particular interest to the

natural history collections community, we found that staining with $I_2$ in 70% EtOH minimizes chemical changes to the fluid with which the specimen interacts, can yield high-quality staining, and allows visualization of mineralized and soft tissues on the same CT scan.

## Supporting information

**S1 Fig. Overlay of the ATR-FTIR spectra of bone samples.** Sample ID numbers: 657966 (unstained control 1; light gray) and 657969 (unstained control 2; light blue), 657963 (stain 3; green), 657968 (stain 2; brown), 657967 (stain 4; purple), and 657965 (stain 1; pink) (normalized to the Amide I peak from collagen at 1646 cm-1).
(TIF)

**S1 Table. Bone sample weights for all conditions.** The six-digit number at the beginning of the sample ID corresponds to the USNM specimen number of the specimen from which the sample was taken. "A" at the end of the number indicates that the sample was taken before staining, "Z" indicates that the sample was taken after refreshing stain, and "B" indicates that the sample was taken once the specimen was fully stained. The "-B" suffix at the end of all the codes denotes bone samples.
(DOCX)

**S2 Table. Muscle sample weights for all conditions.** The six-digit number at the beginning of the sample ID corresponds to the USNM specimen number of the specimen from which the sample was taken. "A" at the end of the number indicates that the sample was taken before staining, "Z" indicates that the sample was taken after refreshing stain, and "B" indicates that the sample was taken once the specimen was fully stained. The "-M" suffix at the end of all the codes denotes muscle samples.
(DOCX)

**S3 Table. MorphoSource ARK IDs for all scans generated by this study.**
(DOCX)

**S1 File. Detailed methods for molecular protocols.**
(DOCX)

**S2 File. Custom R script for calculating the levels of iodination (tyrosine [Y]/histidine [H]), diiodination (Y/H), and triiodination (Y) on the PSM level as quantified by Meta-Morpheus.**
(R)

## Acknowledgments

David Blackburn and anonymous reviewers provided feedback that improved this manuscript. We thank Teresa Feo and Brian Metscher for early discussion of this project. We acknowledge Drew Harper for helpful discussions of the chemistry and properties of stain solutions. CT scan data were generated by Jennifer J. Hill and Scott Whittaker, and specimens were handled by H.F.J. and C.M.M., at the Scientific Imaging Laboratories of the NMNH.

## Author Contributions

**Conceptualization:** Catherine M. Early, Ashley C. Morhardt, Timothy P. Cleland, Christopher M. Milensky, Gwénaëlle M. Kavich, Helen F. James.

**Data curation:** Catherine M. Early, Timothy P. Cleland, Gwénaëlle M. Kavich, Helen F. James.

**Formal analysis:** Timothy P. Cleland, Gwénaëlle M. Kavich.

**Funding acquisition:** Catherine M. Early, Timothy P. Cleland, Gwénaëlle M. Kavich, Helen F. James.

**Investigation:** Catherine M. Early, Ashley C. Morhardt, Timothy P. Cleland, Christopher M. Milensky, Gwénaëlle M. Kavich, Helen F. James.

**Methodology:** Catherine M. Early, Ashley C. Morhardt, Timothy P. Cleland, Christopher M. Milensky, Gwénaëlle M. Kavich.

**Project administration:** Catherine M. Early, Helen F. James.

**Resources:** Timothy P. Cleland, Christopher M. Milensky, Helen F. James.

**Software:** Timothy P. Cleland.

**Supervision:** Catherine M. Early, Helen F. James.

**Visualization:** Catherine M. Early, Ashley C. Morhardt.

**Writing – original draft:** Catherine M. Early, Ashley C. Morhardt, Timothy P. Cleland, Christopher M. Milensky, Gwénaëlle M. Kavich, Helen F. James.

**Writing – review & editing:** Catherine M. Early, Ashley C. Morhardt, Timothy P. Cleland, Christopher M. Milensky, Gwénaëlle M. Kavich, Helen F. James.

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
