## [Decision Letter · Decision Letter 0]

22 May 2020

PONE-D-20-09235

Chemical and molecular effects of diffusion-based iodine contrast-enhancing stains on fluid-preserved avian specimens

PLOS ONE

Dear Dr. Early,

Thank you for submitting your manuscript to PLOS ONE. After careful consideration, we feel that it has merit but does not fully meet PLOS ONE’s publication criteria as it currently stands. Therefore, we invite you to submit a revised version of the manuscript that addresses the points raised during the review process.

The reviewers raised fair number of technical questions for you to address, please revise them one by one with highlight in you revised version.

Please submit your revised manuscript in two weeks. If you will need more time than this to complete your revisions, please reply to this message or contact the journal office at plosone@plos.org. Please include the following items when submitting your revised manuscript:

We look forward to receiving your revised manuscript.

Kind regards,

Jinhui Tao, Ph.D.

Academic Editor

PLOS ONE

Journal Requirements:

Reviewers' comments:

Reviewer's Responses to Questions

**Comments to the Author**

1. Is the manuscript technically sound, and do the data support the conclusions?

Reviewer #1: Yes

Reviewer #2: Yes

Reviewer #3: Partly

2. Has the statistical analysis been performed appropriately and rigorously? 

Reviewer #1: Yes

Reviewer #2: Yes

Reviewer #3: Yes

3. Have the authors made all data underlying the findings in their manuscript fully available?

Reviewer #1: Yes

Reviewer #2: Yes

Reviewer #3: Yes

4. Is the manuscript presented in an intelligible fashion and written in standard English?

Reviewer #1: Yes

Reviewer #2: Yes

Reviewer #3: Yes

5. Review Comments to the Author

Reviewer #1: The present work shows how different iodine-based staining methods affect the staining results and materials of the avian samples in the diceCT imaging. The story and experimental details are clear. However, there are some points that need to be addressed.

1. Page 7, line 147 – 148: In the stepwise dehydration process, would it be better to start from lower EtOH content, for example, 10-20%? Starting from 70% EtOH will dehydrate the biological specimen fast and cause structural damage to the biological tissues.

2. Page 14, line 292-293, figure 4: It is hard to conclude that the soft and flexible tarsometatarsus is caused by the demineralization. The stiffness and hardness of biological tissues (bone, feather, skin…) can be affected by hydration. After stained with I2KI in RO water, the specimen is fully hydrated, leading to the decrease of stiffness and hardness. Just by bending the tarsometatarsus in not enough to verify the demineralization. Also, would it be possible to visualize the demineralization (i.e., pores, damages) in the micro-CT imaging? Is it possible to perform high resolution CT scan on the bones, and see if there are any changes of the structure?

3. Page 18, line 377-380, FTIR analysis. The content of minerals is more related to the intensity ratio, but not the integrated area under the peaks. The relative matrix to mineral ratio should be calculated from the ratio of the intensity of different peaks. The best way to verify and quantify the demineralization is TGA/DSC tests.

4. Page 20, paragraph 2: Again, the “firm” or “flexible” feeling of the sample is also related to the hydration state. In the staining with EtOH, samples are dehydrated completely, thus they will be “firm”. While in samples stained with water solution, samples will be more flexible.

Reviewer #2: The paper is fairly well organized and clear.

Different receipes of the staining were used and compared

on the influence of the sample. Some minor issues addressed would enhance the paper.

1. some minor format problems in the paper, including paragraph format, scale bar in Figure 1 is barely visible,

2. words in the figure 7 is difficult to read

3. I tend to belive that it is an overstatement of the "molecular effects" in terms of the staining on the sample,

the author may need to modify the title of the paper.

Reviewer #3: The authors present an interesting analyses of current staining protocols for diffusible iodine-based contrast-enhanced computed tomography (diceCT) and claims a new recipe with elemental I2 in 70% ethanol that reduces demineralization during the staining process and gives high quality soft-tissue visualization at a shorter time scale. The authors have elucidated the molecular effect of iodine staining to find that amino acid modifications with iodine may be completely or partially irreversible. This study can provide crucial literature for museum curators and researchers looking to use diceCT on rare specimens. Therefore, this article can be accepted after revisions listed below.

Major issues:

1. However, one data set from one bird specimen for staining I2 or I2KI in 70% ethanol is not sufficient to rule out artifacts. Particularly when more protein is observed after staining than before staining for I2 in 70% ethanol. This indicates that protein groups measured before staining and the technique involved could be inaccurate. The author does make a note of this and I am not an expert on preparation of samples for proteomics analysis, but repeat experiments with additional bird specimens can confirm the claims.

2. Repeat experiments are also required for demineralization analysis using FTIR since diffusion based mineral dissolution can often lead to different results if the chemical environment is modified or samples with different chemical compositions, mass or thickness are studied.

3. The authors report that I2 in 70% ethanol did not completely stain the interior of the bird specimen. Since, diceCT is a technique to visualize soft-tissue rather than bone structure which can be obtained without stain, it is imperative that all soft-tissue is adequately stained before any claims for the usefulness of this protocol are published. The authors suggest higher concentration of I2 in 70% ethanol can address this issue, therefore results from these experiments can support their overall claims.

4. Furthermore, the author has pointed out that staining using Lugol’s solution does not cause extensive demineralization in previous reports from literature. However, in the experiments presented in this article, the Lugol’s solution had the highest demineralization and the authors claim that the specimens in this study could have incomplete formalin fixation. This information does question the accuracy of results using ethanol solutions presented in this article. Therefore, the authors might find it useful to look into the protocols used in literature, check if the solutions were buffered at neutral pH and determine extent of fixation before staining.

Minor issues: some typos in lines 39 and 279.

6. PLOS authors have the option to publish the peer review history of their article (what does this mean?). If published, this will include your full peer review and any attached files.

Reviewer #1: No

Reviewer #2: No

Reviewer #3: No

---

## [Author Response · Author response to Decision Letter 0]

5 Jun 2020

We appreciate the feedback provided by the reviewers. We have addressed their comments with our responses here and changes to the manuscript and figures where appropriate.

Reviewer #1: The present work shows how different iodine-based staining methods affect the staining results and materials of the avian samples in the diceCT imaging. The story and experimental details are clear. However, there are some points that need to be addressed.

1. Page 7, line 147 – 148: In the stepwise dehydration process, would it be better to start from lower EtOH content, for example, 10-20%? Starting from 70% EtOH will dehydrate the biological specimen fast and cause structural damage to the biological tissues.

In the second paragraph of the previous “Specimen collection” section, we specify that the specimens were fixed with NBF and then preserved in increasing concentrations of EtOH, ending in 70% EtOH (“We submerged the specimens in NBF for 3 days, rinsed them with water, and then transferred them into graded EtOH concentrations of 25%, 50% and 70% at intervals of three to seven days, following the protocol recommended by [25] and following the staining rationale of [27].”, lines 132-135). We have further clarified this in lines 148-150 so the sentence now reads “All specimens were preserved in 70% EtOH, so the specimen stained with I2 in 100% EtOH was dehydrated stepwise (incubated in 70%, 80%, 90%, 100% EtOH for 48 hr intervals) following established protocols [25,27,28] before staining.”

2. Page 14, line 292-293, figure 4: It is hard to conclude that the soft and flexible tarsometatarsus is caused by the demineralization. The stiffness and hardness of biological tissues (bone, feather, skin…) can be affected by hydration. After stained with I2KI in RO water, the specimen is fully hydrated, leading to the decrease of stiffness and hardness. Just by bending the tarsometatarsus in not enough to verify the demineralization. Also, would it be possible to visualize the demineralization (i.e., pores, damages) in the micro-CT imaging? Is it possible to perform high resolution CT scan on the bones, and see if there are any changes of the structure?

In response to this comment, we revised the paragraph in the Discussion that summarizes our physical condition assessments of the specimens (“Physical condition of specimens”). The last two sentences now read “Compared with an unstained specimen and with the specimens stained in EtOH, the soft tissues of the water-based stained specimens were less firm, the skull vault was soft and flexible, the sternum could no longer be felt by pressing on the trunk, and the long bones of the wing and leg were extremely bendable, especially in the specimen that was stained for ten weeks (Fig 4, Table 2). The softness of the skull vault, loss of a hard sternum, and flexibility of the long bones of the two specimens stained in water-based solutions, after destaining, are physical indications that demineralization had occurred.” This paragraph is now more specific about the changes to the skeleton noted in the specimens stained in a water-based solution, including extreme bendability of long bones, loss of a hard sternum, and softness of the skull vault. 

Although we are grateful for reviewer feedback, we disagree with the point about hydration. In our experience, hydrated long bones of the wing and leg in small birds are stiff and cannot be bent into a half circle without breaking (Fig. 4). Within the living body of the bird, these bones are hydrated but still very stiff, and under bending forces, they break. After dehydrating and rehydrating long bones of small animals, they remain stiff and breakable. We base this on the authorship team's collective experience with preparing skeletal specimens of birds by maceration in water, excavating underwater Holocene fossil sites, and analyzing the histology, proteomics and chemistry of bone samples, which together amounts to about 100 years of experience working with animal bones. We note that there are bones in the avian skeleton that are more flexible than the major long bones in the living bird, such as ribs and often the furcula. Although our study does not refer to those specific bones, they also cannot be bent into a half circle without breaking after dehydration and rehydration. 

In the paper, we present multiple lines of evidence supporting demineralization of the skeleton, including the physical condition assessments, the FTIR analysis, the data on the increasing acidity of the water-based solution during staining, and the relatively low radiopacity of the skeleton in CT scans of specimens treated with water-based stain. We have deleted the second sentence of the Discussion, which was misleading because it mentioned only physical assessments as evidence of demineralization. Later in the paragraph, we mention all the sources of evidence in one sentence. Unfortunately, we do not have access to a CT scanner capable of yielding the resolution necessary to assess the microstructural changes that the reviewer asks about.

3. Page 18, line 377-380, FTIR analysis. The content of minerals is more related to the intensity ratio, but not the integrated area under the peaks. The relative matrix to mineral ratio should be calculated from the ratio of the intensity of different peaks. The best way to verify and quantify the demineralization is TGA/DSC tests.

We have retained integrated area for this estimation based on multiple usages of integrated area for FT-IR in the literature (Paschalis et al., 2017, Gourion-Arsiquaud et al., 2009, Pienkowski et al., 1997, Taylor et al., 2017). It is much more common to apply intensity calculations for Raman data. We agree that TGA/DSC tests could be useful, but we were limited in material to the contralaterally controlled samples preventing us from performing additional destructive sampling.

We have added the following to the ATR-FTIR methods section: “To avoid destructive sampling (e.g., using TGA/DSC tests) of limited bone samples, we used ATR-FTIR to estimate mineral-to-matrix ratios.”

Paschalis, E. P., Gamsjaeger, S., & Klaushofer, K. (2017). Vibrational spectroscopic techniques to assess bone quality. Osteoporosis International, 28(8), 2275-2291.

Gourion‐Arsiquaud, S., Burket, J. C., Havill, L. M., DiCarlo, E., Doty, S. B., Mendelsohn, R., ... & Boskey, A. L. (2009). Spatial variation in osteonal bone properties relative to tissue and animal age. Journal of Bone and Mineral Research, 24(7), 1271-1281.

Pienkowski, D., Doers, T. M., Monier‐Faugere, M. C., Geng, Z., Camacho, N. P., Boskey, A. L., & Malluche, H. H. (1997). Calcitonin alters bone quality in beagle dogs. Journal of Bone and Mineral Research, 12(11), 1936-1943.

Taylor, E. A., Lloyd, A. A., Salazar-Lara, C., & Donnelly, E. (2017). Raman and Fourier transform infrared (FT-IR) mineral to matrix ratios correlate with physical chemical properties of model compounds and native bone tissue. Applied spectroscopy, 71(10), 2404-2410.

4. Page 20, paragraph 2: Again, the “firm” or “flexible” feeling of the sample is also related to the hydration state. In the staining with EtOH, samples are dehydrated completely, thus they will be “firm”. While in samples stained with water solution, samples will be more flexible.

Please see our response to the reviewer's second comment.

Reviewer #2: The paper is fairly well organized and clear.

Different recipes of the staining were used and compared on the influence of the sample. Some minor issues addressed would enhance the paper.

1. some minor format problems in the paper, including paragraph format, scale bar in Figure 1 is barely visible,

We are unsure what the reviewer means by problems with paragraph format and so we were unable to address this comment. If you can provide specific examples, we are happy to consider changes. We have increased the size of the scale bars in Figure 1. 

2. words in the figure 7 is difficult to read

We have increased the font size of the text in Figure 7. 

3. I tend to believe that it is an overstatement of the "molecular effects" in terms of the staining on the sample, the author may need to modify the title of the paper.

We have changed the title to “Chemical effects of diceCT staining protocols on fluid-preserved avian specimens.”

Reviewer #3: The authors present an interesting analyses of current staining protocols for diffusible iodine-based contrast-enhanced computed tomography (diceCT) and claims a new recipe with elemental I2 in 70% ethanol that reduces demineralization during the staining process and gives high quality soft-tissue visualization at a shorter time scale. The authors have elucidated the molecular effect of iodine staining to find that amino acid modifications with iodine may be completely or partially irreversible. This study can provide crucial literature for museum curators and researchers looking to use diceCT on rare specimens. Therefore, this article can be accepted after revisions listed below.

Major issues:

1. However, one data set from one bird specimen for staining I2 or I2KI in 70% ethanol is not sufficient to rule out artifacts. Particularly when more protein is observed after staining than before staining for I2 in 70% ethanol. This indicates that protein groups measured before staining and the technique involved could be inaccurate. The author does make a note of this and I am not an expert on preparation of samples for proteomics analysis, but repeat experiments with additional bird specimens can confirm the claims.

We agree that additional testing with other specimens would be beneficial, but in this comparison paper it provides a useful data point without additional testing at this time. Additionally, the stochastic nature of how the data were collected could also explain the difference in identifications between the pre- and post-treatment samples. Additional testing is beyond the scope of this study, but could be applied in similar future studies.

We have added the following: “The increases in identifications could be derived from stochastic differences in data acquisition, but future studies with additional samples will clarify if this increase in the number of protein groups is intrinsic to staining with I2 in 70% EtOH.”

2. Repeat experiments are also required for demineralization analysis using FTIR since diffusion based mineral dissolution can often lead to different results if the chemical environment is modified or samples with different chemical compositions, mass or thickness are studied.

Because of our contralateral sampling procedure, we are limited to one bone pre- and post-treatment. We agree that bones of different thicknesses or locations may have different levels of demineralization based on diffusion, but other lines of evidence we have support that our current FT-IR data is a reasonable reflection of what is occurring (especially in the Lugol’s iodine sample). We were able to qualitatively detect that skull, breast, wings, and legs were flexible after the staining procedure. Additionally, the relative intensity of the bone on the microCT data is diminished (suggesting demineralization is occurring) compared to the adjacent stained tissues. The ethanol birds show greater contrast in the bones even with staining. 

3. The authors report that I2 in 70% ethanol did not completely stain the interior of the bird specimen. Since, diceCT is a technique to visualize soft-tissue rather than bone structure which can be obtained without stain, it is imperative that all soft-tissue is adequately stained before any claims for the usefulness of this protocol are published. The authors suggest higher concentration of I2 in 70% ethanol can address this issue, therefore results from these experiments can support their overall claims.

We have added a line at the end of the “Quality of differential staining” section: “Future studies should attempt to improve penetration of I2 in 70% EtOH to the center of specimens through higher stain concentration, longer staining times, or refreshing of the solution throughout staining.”

4. Furthermore, the author has pointed out that staining using Lugol’s solution does not cause extensive demineralization in previous reports from literature. However, in the experiments presented in this article, the Lugol’s solution had the highest demineralization and the authors claim that the specimens in this study could have incomplete formalin fixation. This information does question the accuracy of results using ethanol solutions presented in this article. Therefore, the authors might find it useful to look into the protocols used in literature, check if the solutions were buffered at neutral pH and determine extent of fixation before staining.

We are intimately familiar with the diceCT protocols used in the literature. The use of buffered staining solutions is not widespread in the diceCT community, likely due in part to the fact that the effects we observed have not been observed or directly tested for in many previous studies. After conferring with museum conservators and experts on fluid fixation of vertebrate specimens, we have confirmed our understanding that “extent of fixation” is a subjective criterion and cannot be quantitatively determined.

We adjusted our mention of incomplete fixation because we realized it was speculation and resulted in confusion. Our purpose in bringing up fixation was to highlight the risk to other museum specimens with unknown or variable preservation histories, but our specimens have neither of these. We have modified this section to the following: “We suggest that the extended time period in acid, water-based stain is leading to these changes. We also hypothesize that the degree of fixation, which was complete and standardized in our specimens but is known to vary in museum specimens, may influence the susceptibility of a specimen to these changes.”

Minor issues: some typos in lines 39 and 279.

Thank you for catching those. We have corrected them.

---

## [Decision Letter · Decision Letter 1]

25 Aug 2020

Chemical effects of diceCT staining protocols on fluid-preserved avian specimens

PONE-D-20-09235R1

Dear Dr. Early,

We’re pleased to inform you that your manuscript has been judged scientifically suitable for publication and will be formally accepted for publication once it meets all outstanding technical requirements.

Kind regards,

Jinhui Tao, Ph.D.

Academic Editor

PLOS ONE

Additional Editor Comments (optional):

As this is a comparison paper which points out the molecular effects of iodine staining on biological samples, it already provides a useful data point without additional testing at this time. Additionally, the stochastic nature of how the data were collected could also explain the difference in identifications between the pre- and post-treatment samples. I do agree that additional testing of a second bird sample is beyond the scope of this study.

Reviewers' comments:

Reviewer's Responses to Questions

**Comments to the Author**

1. If the authors have adequately addressed your comments raised in a previous round of review and you feel that this manuscript is now acceptable for publication, you may indicate that here to bypass the “Comments to the Author” section, enter your conflict of interest statement in the “Confidential to Editor” section, and submit your "Accept" recommendation.

Reviewer #1: All comments have been addressed

Reviewer #2: (No Response)

Reviewer #3: (No Response)

2. Is the manuscript technically sound, and do the data support the conclusions?

Reviewer #1: Yes

Reviewer #2: (No Response)

Reviewer #3: Partly

3. Has the statistical analysis been performed appropriately and rigorously? 

Reviewer #1: N/A

Reviewer #2: (No Response)

Reviewer #3: N/A

4. Have the authors made all data underlying the findings in their manuscript fully available?

Reviewer #1: Yes

Reviewer #2: (No Response)

Reviewer #3: Yes

5. Is the manuscript presented in an intelligible fashion and written in standard English?

Reviewer #1: Yes

Reviewer #2: (No Response)

Reviewer #3: Yes

6. Review Comments to the Author

Reviewer #1: The authors have addressed my comments adequately in the revision. The paper can be published in its current form.

Reviewer #2: (No Response)

Reviewer #3: (No Response)

7. PLOS authors have the option to publish the peer review history of their article (what does this mean?). If published, this will include your full peer review and any attached files.

Reviewer #1: No

Reviewer #2: No

Reviewer #3: No

---

## [Editor Report · Acceptance letter]

1 Sep 2020

PONE-D-20-09235R1 

Chemical effects of diceCT staining protocols on fluid-preserved avian specimens 

Dear Dr. Early:

I'm pleased to inform you that your manuscript has been deemed suitable for publication in PLOS ONE. Congratulations! Your manuscript is now with our production department. 

Kind regards, 

on behalf of

Dr. Jinhui Tao 

Academic Editor

PLOS ONE